# Prior Antibiotic Use Increases Risk of Urinary Tract Infections Caused by Resistant *Escherichia coli* among Elderly in Primary Care: A Case-Control Study

**DOI:** 10.3390/antibiotics11101382

**Published:** 2022-10-09

**Authors:** Maria L. V. Jensen, Volkert Siersma, Lillian M. Søes, Dagny Nicolaisdottir, Lars Bjerrum, Barbara J. Holzknecht

**Affiliations:** 1Department of Clinical Microbiology, Copenhagen University Hospital—Herlev and Gentofte, 2730 Herlev, Denmark; 2The Research Unit for General Practice and Section of General Practice, Department of Public Health, University of Copenhagen, 1014 Copenhagen, Denmark; 3Department of Clinical Microbiology, Copenhagen University Hospital—Hvidovre and Amager, 2650 Hvidovre, Denmark; 4Department of Clinical Medicine, University of Copenhagen, 2200 Copenhagen, Denmark

**Keywords:** antibiotic prescribing, antimicrobial resistance, elderly, general practice, urinary tract infection, *Escherichia coli*

## Abstract

We investigated whether prior use of antibiotics affects the risk of mecillinam/trimethoprim/nitrofurantoin/multi-resistant *Escherichia coli* urinary tract infection (UTI) among elderly patients in general practice. Data on urine culture came from urine samples performed in general practice and sent to hospitals in the Capital Region of Denmark, and prescription data came from a nationwide prescription database. The study population consisted of patients with UTI episodes (*n* = 41,027) caused by *E. coli* that received a concurrent antibiotic prescription against UTI from 2012 to 2017. We used a case-control design. Cases were UTI episodes caused by mecillinam, trimethoprim, nitrofurantoin or multi-resistant *E. coli* and controls were UTI episodes caused by *E. coli* not displaying the respective resistance pattern. We analyzed whether exposure to antibiotics in a period of 8–90 days prior to the UTI episode affected the risk of antibiotic resistant uropathogenic *E coli*. The analyses were adjusted for age, sex, hospital admission and nursing home status. The odds of resistance to all of the four antibiotics increased significantly after exposure to antibiotics within 90 days prior to the UTI episode. In general, mecillinam showed the lowest increase in the odds for selection of resistance. The results indicate that mecillinam is a favorable antibiotic choice in terms of selection of resistance.

## 1. Introduction

Antimicrobial resistance (AMR) is a major threat to global health [1] and increased antimicrobial consumption is associated with increased resistance rates [2]. The majority of antibiotic treatments are prescribed by general practitioners (GPs) [3], and urinary tract infection (UTI) is one of the most frequent indications in primary care. The most common causative agent of UTI is *Escherichia coli* and infections are often treated empirically. However, a high rate of antimicrobial resistance is increasingly causing treatment failure [4].

Prior studies have shown associations between antibiotic use and AMR at both the individual level [5,6,7] and the population level [2]. For UTI, these associations have been shown for common UTI agents, such as trimethoprim, nitrofurantoin and fluoroquinolones. A dose–response relationship has also been illustrated [8,9]. Most studies have indicated a critical 3-months post-exposure period of increased risk of infection with resistant pathogens in the clinical setting at an individual level [8,10,11], but others have indicated that a longer period should be taken into account [6].

In Denmark, pivmecillinam (the prodrug of mecillinam) is first line treatment for UTI in general practice and, to our knowledge, no studies have examined the association between prior antibiotic use and risk of UTI caused by mecillinam resistant *E. coli* based on person-specific data. Nitrofurantoin and trimethoprim are second line choices in case of penicillin allergy, and both are frequently used in other European countries as a first line treatment. In Denmark, the resistance rate of the three agents for *E. coli* in urine samples from general practices differs: mecillinam 4.9%, trimethoprim 21% and nitrofurantoin 0.8% [12].

The elderly population is highly relevant to the study in relation to AMR as it is a high antibiotic-exposure population [12,13,14]. This population furthermore has a high rate of hospital admissions and residency at long term care facilities, which both are risk environments for colonization with antimicrobial resistant bacteria and antibiotic treatment [15,16,17,18,19].

To avoid unnecessary antibiotic use and at the same time limit the risk of treatment failure, it is important for the GP to be able to identify individuals at high risk of AMR. We set out to investigate whether prior outpatient exposure to antibiotics affects the risk of mecillinam/trimethoprim/nitrofurantoin/multi-resistant *E. coli* in UTI among elderly patients in general practice.

## 2. Materials and Methods

This series of case-control studies use data on all urine samples from elderly people (aged ≥ 65 years) in general practice with significant growth of *E. coli*, submitted to the Departments of Clinical Microbiology (DCM) in hospitals at the Capital Region of Copenhagen from 2012 to 2017, combined with data on antibiotic prescriptions. The study only included urine samples with significant growth of *E. coli* and susceptibility testing to one or more of the following antibiotics: mecillinam, nitrofurantoin or trimethoprim. In order to exclude episodes of asymptomatic bacteriuria (ASB), we only included urine samples where a UTI treatment had been redeemed within two days before and five days after the date the urine sample was taken (hereafter referred to as a UTI episode). UTI treatments were defined as prescription of the following antibiotics available for UTI treatment in Denmark: pivmecillinam, trimethoprim, nitrofurantoin, sulfamethizole, ciprofloxacin, pivampicillin and amoxicillin. A UTI episode was included in the analysis if a full exposure period was available, i.e., there was at least a 90-day interval between two UTI episodes. Figure 1 visualizes the timeline of the study.

### 2.1. Data Sources

Data on the urine samples (mainly midstream urine samples and urine sampled via catheter) were extracted from the two DCMs’ laboratory information systems. Data included sampling date and culture results, including susceptibility testing results.

Antibiotic exposure data were retrieved from the Danish National Prescription Registry (DNPR). This registry contains individual level information on all prescriptions redeemed at a pharmacy since 1995 and has high validity and completeness [20]. In Denmark, antibiotics are exclusively available by prescription and pharmacies have monopoly on sale. All sales linked to the individual, i.e., prescription medicines not administered during hospital admission, are captured in the DNPR.

Data on hospital admission history were collected from the Danish national patient registry. This registry contains administrative data on all contacts in the secondary health care sector, such as date of admission, type of admission and diagnoses related to the admission. Data on age and sex were provided by Statistics Denmark. The different data sources were linked by the unique 10-digit personal identification number.

### 2.2. Laboratory Methods

Urinalysis was performed according to European guidelines [21]. Urine specimens were plated on split plates (5% horse blood and a chromogenic UTI-agar) and read after overnight incubation at 35 ± 1 °C in atmospheric air. Significant growth of *E. coli* was defined as minimum 1000 CFU/mL. Bacterial species identification was based on phenotypical and biochemical characteristics and possibly confirmed using Matrix-Assisted Laser Desorption-Ionization time of flight mass spectrometry. Susceptibility testing was performed using EUCAST disk diffusion methodology and breakpoints [22]. For trimethoprim, the breakpoints implemented per 1 January 2020 (R < 15 mm to S ≥ 15 mm) were applied throughout the study period, meaning that former “I = intermediate” was categorized as “S = susceptible”.

### 2.3. Outcome Measure and Classification of Cases and Controls

Four parallel case-control analyses were performed, one for each type of resistance. This means that for each analysis all UTI episodes were classified as either case or control depending on the result of the susceptibility report. The four outcomes were resistance to mecillinam, trimethoprim, nitrofurantoin, and multi-resistance, the latter defined as resistance to all three agents at the same time.

Cases were defined as UTI episodes with significant growth of mecillinam or trimethoprim or nitrofurantoin or multi-resistant *E. coli.* All remaining UTI episodes, i.e., samples with significant growth of *E. coli* susceptible to mecillinam or trimethoprim or nitrofurantoin or all three simultaneously, were defined as the corresponding controls. If more than one *E. coli* isolate was identified in a urine sample, resistance overruled susceptibility.

### 2.4. Antibiotic Exposure

Outpatient antibiotic exposure was defined as a prescription of an antibiotic agent according to the Anatomical Therapeutic Chemical Classification System, ATC code J01: antibiotics for systematic use and P01AB01: metronidazole. The exposure to prior antibiotic use was measured from eight days before the sampling date of the urine sample to 90 days before the sample date. Exposure in the 0–7 days prior to the sample date was not included in the exposure as this most likely reflects treatment failure.

Prior antibiotic exposure was measured in the following ways: (1) total exposure (i.e., number of prescriptions and number of defined daily doses (DDD)). Number of prescriptions were categorized as no prescriptions, 1 prescription, 2 prescriptions or ≥3 prescriptions. DDD was grouped as no exposure and three equally large groups of exposure constructed as follows: >0–33.3 percentile, >33.3–66.6 percentile and >66.6 percentile. All specific numbers are provided as footnotes in the tables; (2) Time since last prescription (8–30 days, 31–60 days, 61–90 days or no exposure within 8–90 days); and (3) exposure to different specific types of antibiotics (phenoxymethylpenicillin and dicloxacillin, pivmecillinam, amoxicillin, amoxicillin + β-lactamase inhibitor, trimethoprim, sulfamethizole, nitrofurantoin, macrolides, quinolones, others, or no exposure within 8–90 days).

### 2.5. Covariates

For the covariate hospital admission both the number of hospital admissions and the number of days admitted to hospital in the exposure period were collected. Only full-day admissions were included. The number of hospital admissions were grouped as 0 admissions, 1 admission, and >1 admission within the exposure period. The number of days admitted to hospital were grouped as 0 days, 1–7 days, and >7 days. Furthermore, the number of admissions and the number of days of admissions related to infectious diseases were calculated based on the recorded diagnosis of the admission using ICD-10. A list of all included diagnoses is available in the Appendix A. We included year of analysis due to changes in AMR rates over time, hence we adjusted the analysis for the year the sample was analyzed. The two DCMs service around 50% of the GPs each but differ in sample loads and potentially in sampling characteristics, e.g., whether a primary microbiological susceptibility testing is performed by the GP for uncomplicated cases or sent to the laboratory for susceptibility testing. We therefore adjusted the analysis for the laboratory that analyzed the urine sample. Furthermore, resistance rates may differ in different geographical areas. Lastly, nursing home status was included. To identify individuals likely living in nursing homes we defined an address as a nursing home if ≥3 individuals aged 80 years or older in the study population are registered at that same address.

### 2.6. Statistical Analysis

The data for this study were analyzed in a series of case-control designs. The association between prior use of antibiotics and resistance was analyzed with odds ratios (ORs) and 95% confidence intervals (CI95%) from multivariable logistic regression models. The unit of analysis was UTI episodes. Analyses were performed separately for all outcomes and exposure measures. Because of the possible multiple UTI episodes per person, the variance were adjusted, and thereby the confidence interval of the OR, for this excess dependence, by the method of generalized estimating equations [23]. To adjust for potential confounding, the following covariates were additionally included in the models: age, sex, hospital admission history, nursing home residency, which laboratory analyzed the urine sample and the year the sample was analyzed. The results of the unadjusted analysis are presented in Appendix A.

### 2.7. Sensitivity Analysis

An individual with several episodes of UTI could be included more than once if there was a full exposure period of 90 days between the episodes. In a sensitivity analysis, the analyses described above in a dataset were repeated by selecting a single random UTI episode from each individual.

## 3. Results

### 3.1. General Characteristics

Within the study period, we identified 51,779 UTI episodes, defined as urine samples with significant growth of *E. coli* and concurrent antibiotic prescription for UTI. After exclusion of 10,752 urine samples, where a full exposure period was not available (i.e., there was less than 90 days to the previous UTI episode), 41,027 UTI episodes were included in the study. The inclusion of urine samples is shown in Figure 2 below. The median age of the patients included in the study was 78 years (interquartile range: 71–85) and 84% were female. Baseline characteristics are provided in Table 1. For each analysis we excluded a number of observations due to missing information regarding susceptibility for the specific antibiotic in question; mecillinam: 91, trimethoprim: 71, nitrofurantoin: 2570. For multi-resistance 14,101 samples were not fully susceptible or had missing information of the susceptibility pattern for one or more of the three antibiotics and were therefore excluded from the analysis. In our study material we found the following resistance rates: mecillinam 4.5%, trimethoprim 26.3%, nitrofurantoin 2.3%, and multi-resistance 0.3%.

### 3.2. Total Exposure

The analyses showed that the odds of resistance increased significantly after exposure to antibiotic. There was a dose-response correlation with both the number of antibiotic prescriptions and the number of DDDs. The increase in odds for mecillinam resistance were generally lower than for the other types of resistance, especially for the groups with the highest exposure to antibiotics, i.e., three or more prescriptions and above the 66.6 percentile of DDDs. All results are presented in Table 2.

### 3.3. Time since Last Antibiotic Exposure

For all four types of resistance, the odds of resistance increased with decreasing time since the last antibiotic exposure. The odds for mecillinam resistance increased 1.51 (1.27;2.13) times when exposed 61–90 days prior to the UTI episode and 2.13 (1.91;2.38) times when exposed 8–30 days prior to the UTI episode compared to no exposure. For trimethoprim and nitrofurantoin, the corresponding ORs were 2.20 (2.09;2.32) and 2.83 (2.38;3.37). For multi-resistance, the estimate was even higher. The OR was 5.81 (3.37;10.03) for exposure within 8–30 days. All results are presented in Table 2.

### 3.4. Specific Antibiotic Exposure

The results of the adjusted analysis of exposure to specific antibiotic agents and odds of resistance are shown in Table 3. For mecillinam, exposure to almost all agents increased the odds of resistance significantly. Exposure to phenoxymethylpenicillin or dicloxacillin was associated with a nearly two-fold (OR: 2.01 (1.75;2.31)) increase of mecillinam resistance while exposure to pivmecillinam was associated with a lower increase (OR: 1.62 (1.46;1.80)) of mecillinam resistance.

For trimethoprim and nitrofurantoin, exposure to the agent itself increased the odds considerably more than exposure to any other agent. For trimethoprim, the odds increased 6.48 (5.93;7.09) times and for nitrofurantoin the odds increased 8.64 (7.26;10.39) times. For trimethoprim, exposure to all other agents also increased the odds for resistance significantly. Exposure to quinolones significantly increased the odds for resistance both for trimethoprim and nitrofurantoin.

For multi-resistance, several agents increased the odds significantly. Exposure to amoxicillin + β-lactamase inhibitor, increased the odds 10.18 (4.62;16.26) times, while exposure to trimethoprim had an OR of 8.67 (4.62;16.26) compared to no exposure. Exposure to phenoxymethylpenicillin or dicloxacillin, sulfamethizole and quinolones did not significantly increase the odds for multi-resistance.

### 3.5. Sensitivity Analysis

The results of the sensitivity analysis (not included in the tables) where individuals could only be included once, and the included sample was chosen randomly showed similar but slightly higher estimates than the main analysis. The results are presented in the Appendix A.

## 4. Discussion

The results of the study support current evidence that prior exposure to antibiotics is a risk factor for resistance at the individual patient level. In general, the odds increased with increasing numbers of prescriptions and DDDs and with decreasing time between last antibiotic exposure and the UTI episode.

To our knowledge, this is the first study to examine how prior use of antibiotics affects the risk of UTI caused by mecillinam resistant *E. coli* using individual level data. The results illustrate a clear association between prior outpatient antibiotic use and increased odds for mecillinam resistance in the elderly population. However, the estimates were generally lower than for trimethoprim and nitrofurantoin. Pivmecillinam is mainly used in the Nordic countries but is an interesting option as resistance levels have remained low, despite high use for many years [12]. In the elderly population it is the most commonly prescribed antibiotic agent [24].

In line with our findings, Opatowski et al. [8] found that prior use of antibiotics is associated with increased odds of UTI with antibiotic resistant bacteria. As in our study, Opatowski et al. used register data and collected a large number of cases and controls. Furthermore, they also excluded exposure to antibiotics the week prior to the date of the urine sample. Despite the similarities of data sources, they sampled based on incident hospital admissions for UTI, did not exclusively study the elderly population and included a wider range of pathogens. The study populations therefore differ. Additionally, the AMR rate is higher in France than in Denmark [25], which implies a higher a priori risk of AMR than in the Danish setting. Individuals admitted to hospital due to UTI may also be at an even higher risk of infections caused by antibiotic resistant bacteria compared to our study population, where the urine samples were provided from general practice. Despite these differences, both studies support that prior use of antibiotics increases the risk of resistance in a three-month period post exposure.

While some studies have examined the risk of prior mecillinam use, these studies have focused on outcomes such as Extended-Spectrum β-Lactamase (ESBL) *E. coli* [26,27] and not specifically on mecillinam resistance. However, ESBL-*E. coli* only represents a minor segment of resistance in urinary *E. coli* isolates and these studies therefore only provide a partial picture of the selection of AMR. Mecillinam has been used in the Nordic countries for many years and studies on the relation between mecillinam use and mecillinam resistance are warranted. Our study shows that the use of mecillinam does not drive resistance against itself at the same level as the other antibiotics recommended for UTI in Denmark (trimethoprim and nitrofurantoin). Despite the high use of mecillinam in the past two decades in Denmark, the DANMAP report [12] has shown a stable rate of resistance to mecillinam in *E. coli* in urinary samples from general practice. A German study likewise showed a low rate of mecillinam resistance [28]. Furthermore, another study showed a high genetic diversity of *E. coli* resistant to mecillinam, indicating no or little clonal spread of mecillinam resistance [29]. This, together with the pharmacokinetic and dynamic advantages of mecillinam (e.g., β-lactam antibiotic, bactericidal effect), indicates that mecillinam is a favorable choice of treatment among the elderly population infected with *E. coli* UTI in Denmark in terms of selection.

Bell et al. [5] and Costello et al. [6] reviewed the literature of prior antibiotic use and risk of resistance, and both found an association between antibiotic use and AMR. This is in line with the most recent review published by Bakhit et al. [7] in 2018. Generally, a decay was observed within the first month, but for some pathogens the resistance persisted up to three months. Overall, the authors concluded that long term follow-up was lacking. Our study shows a significant increase in odds for resistance when exposed to antibiotic up to three months prior to the sample date (except for multi-resistance where the association was only significant for 60 days prior to the sample date). As we did not study beyond three months prior to the sample date, our results can only support a three month-window of increased risk. However, we cannot conclude whether the risk extends beyond the three months tested in this study.

The main strength of this study is the large number of cases and controls included and the data sources applied. As antibiotics are exclusively available by prescription in Denmark, the DNPR captures outpatient antibiotic accurately. Furthermore, data on microbiological testing were provided by the DCMs, where the culturing and subsequent susceptibility testing rely on international guidelines. Because of the retrospective nature of the study design using routine data from microbiology reports, we did not have information on clonality or other characterization of the resistant *E. coli* isolates. To increase the probability of sampling incident UTI episodes and not ASB, we only included urine samples where the patient had redeemed a UTI treatment concurrently. However, some GPs may still treat ASB with antibiotics and we cannot exclude that some cases of ASB have been included. This is a limitation of the study.

Further limitations of the study include that we chose to only include UTI episodes caused by *E. coli.* UTI can be caused by a variety of different pathogens, but *E. coli* is responsible for the vast majority of all UTIs among the elderly. We did not include all potential uropathogens, as it is difficult to address intrinsic resistance in this study design and there are no internationally recognized breakpoints for all potential uropathogens. Thus, the external validity of the study is limited to including individuals infected with *E. coli*. Furthermore, our study population was obtained from urine samples analyzed at the DCMs, and our results should therefore only be extended to elderly patients where the GP has performed in-house microbiological testing, very cautiously, if at all. Despite the DNPR having high validity and completeness, we have no knowledge about compliance with redeemed prescriptions or self-medication. Hence, we cannot exclude that some misclassification of exposure has occurred. Another limitation is that we have no information on antibiotic use in the hospital care sector, as antibiotics used in this setting are not captured in the DNPR. However, we adjusted the analysis for hospital admission, and hence sought to limit this potential misclassification bias as much as possible. Finally, we did not have clinical information such as travel history and animal husbandry, which affect resistance prevalence.

The findings in this study support current evidence of the prior use of antibiotics as a risk factor for selection of resistance. Furthermore, it underlines the importance of studying the elderly population. The elderly population is a high-exposure population and could potentially act as a reservoir for AMR. The study also indicates that mecillinam is a favorable choice for the treatment of UTI caused by *E. coli* as it does not select for mecillinam resistance at the same level as trimethoprim and nitrofurantoin select for resistance to trimethoprim and nitrofurantoin, respectively.

## Figures and Tables

**Figure 1 antibiotics-11-01382-f001:**
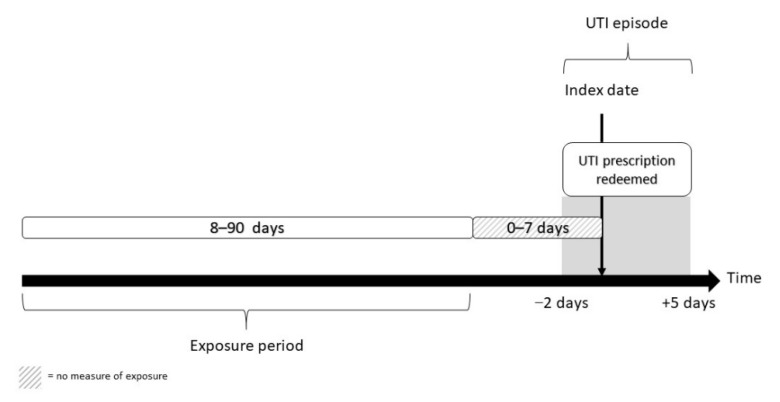
Inclusion of UTI episodes and ascertainment of exposure.

**Figure 2 antibiotics-11-01382-f002:**
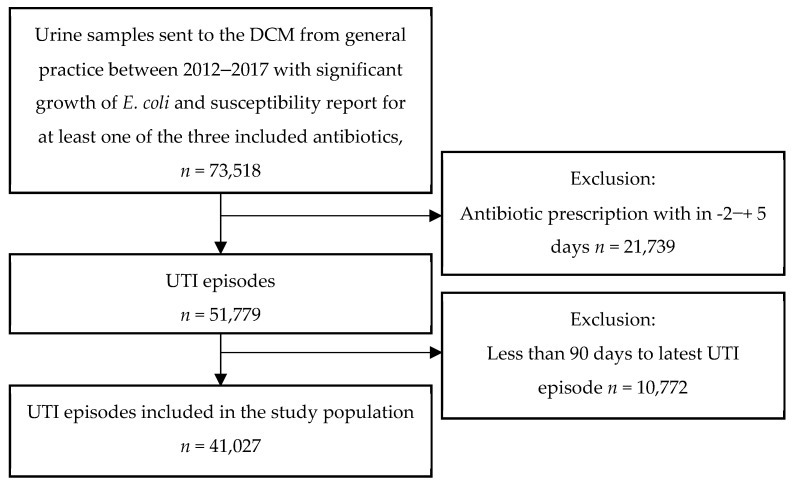
The inclusion of urine samples in the study.

**Table 1 antibiotics-11-01382-t001:** Baseline characteristics.

		Mecillinam		Trimethoprim		Nitrofurantoin		Multi-resistance	
		Cases	Controls	Cases	Controls	Cases	Controls	Cases	Control
		*n* = 1853	*n* = 39,083	*n* = 10,767	*n* = 30,189	*n* = 886	*n* = 37,571	*n* = 72	*n* = 26,854
Age [Median, (IQR *)]		80 (73–87)	78 (71–85)	82 (75–88)	78 (71–85)	82 (75–88)	78 (71–85)	83 (76–90)	78 (71–85)
Female (%)		80	84	83	84	72	84	74	84
Time since last prescription [*n* (%)]	No exposure	715 (38.6)	22,055 (56.4)	4451 (41.3)	18,324 (60.7)	251 (28.3)	21,025 (56.0)	16 (22.2)	16,463 (61.3)
	8–30 days	708 (38.2)	9204 (23.6)	3784 (35.1)	6137 (20.3)	388 (43.8)	8999 (24.0)	37 (51.4)	5310 (19.8)
	31–60 days	262 (14.1)	4681 (12.0)	1585 (14.7)	3362 (11.1)	170 (19.2)	4496 (12.0)	14 (19.4)	2967 (11.1)
	61–90 days	168 (9.1)	3143 (8.0)	947 (8.8)	2366 (7.8)	77 (8.7)	3051 (8.1)	5 (6.9)	2114 (7.9)
Number of prescriptions [*n* (%)]	0	715 (38.6)	22,055 (56.4)	4451 (41.3)	18,324 (60.7)	251 (28.3)	21,025 (56.0)	16 (22.2)	16,463 (61.3)
	1	494 (26.7)	9170 (23.5)	2750 (25.5)	6916 (22.9)	215 (24.3)	8899 (23.7)	11 (15.3)	6137 (22.9)
	2	286 (15.4)	4258 (10.9)	1623 (15.1)	2928 (9.7)	159 (18.0)	4114 (11.0)	16 (22.2)	2542 (9.5)
	≥3	358 (19.3)	3600 (9.2)	1943 (18.1)	2021 (6.7)	261 (29.5)	3533 (9.4)	29 (40.3)	1712 (6.4)
Number of DDD [*n* (%)]	0	715 (38.6)	22,055 (56.4)	4451 (41.3)	18,324 (60.7)	251 (28.3)	21,025 (56.0)	16 (22.2)	16,463 (61.3)
	>0–33.3 percentile **	261 (14.1)	5800 (14.8)	1594 (14.8)	4470 (14.8)	89 (10.1)	5624 (15.0)	6 (8.3)	4004 (14.9)
	>33.3–66.6 percentile	362 (19.6)	5678 (14.5)	1820 (16.9)	4226 (14.0)	151 (17.0)	5547 (14.8)	11 (15.3)	3723 (13.9)
	>66.6 percentile	515 (27.8)	5550 (14.2)	2902 (27.0)	3169 (10.5)	395 (44.6)	5375 (14.3)	39 (54.2)	2664 (9.9)
Exposure within 90 days to the following agents [*n* (%)]									
Phenoxymethylpenicillin or dicloxacillin		279 (15.1)	2899 (7.4)	964 (9.0)	2219 (7.4)	87 (9.8)	2928 (7.8)	6 (8.3)	1916 (7.1)
Pivmecillinam		631 (34.1)	8705 (22.3)	3088 (28.7)	6251 (20.7)	305 (34.4)	8494 (22.6)	37 (51.4)	5487 (20.4)
Amoxicillin		95 (95.1)	986 (2.5)	353 (3.3)	728 (2.4)	18 (2.0)	998 (2.7)	- ***	-
Amoxicillin + β lactamase inhibitor		43 (2.3)	428 (1.1)	165 (1.5)	306 (1.0)	16 (1.8)	430 (1.1)	7 (9.7)	258 (1.0)
Trimethoprim		155 (8.4)	2157 (5.5)	1673 (15.5)	640 (2.1)	85 (9.6)	2130 (5.7)	16 (22.2)	575 (2.1)
Sulfamethizole		176 (9.5)	3001 (7.7)	1081 (10.0)	2102 (7.0)	80 (9.0)	2954 (7.9)	7 (9.72)	1892 (7.1)
Nitrofurantoin		187 (10.1)	2323 (5.9)	1118 (10.4)	1392 (4.6)	356 (40.2)	2023 (5.4)	23 (31.9)	1090 (4.1)
Macrolides		47 (2.5)	1087 (2.8)	350 (3.3)	782 (2.6)	23 (2.6)	1036 (2.8)	-	-
Quinolones		70 (3.8)	1194 (3.1)	644 (6.0)	621 (2.1)	68 (7.7)	1156 (3.1)	5 (6.9)	551 (2.1)
Others		185 (10.0)	2004 (5.1)	837 (7.8)	1358 (4.5)	74 (8.4)	2012 (5.4)	11 (15.3)	1183 (4.4)
Number of admissions [*n* (%)]	0 ****	1344 (72.5)	31,970 (81.8)	8452 (78.5)	24,870 (82.4)	686 (77.4)	30,612 (81.4)	57 (79.2)	22,218 (82.7)
	1	249 (13.4)	4271 (10.9)	1346 (12.5)	3179 (10.5)	114 (12.9)	4117 (11.0)	8 (11.1)	2786 (10.4)
	≥2	260 (14.0)	2842 (7.3)	969 (9)	2140 (7.1)	86 (9.7)	2842 (7.6)	7 (9.7)	1850 (6.9)
Number of admission days [*n* (%)]	0 ****	1393 (75.2)	32,904 (84.2)	8718 (81.0)	25,585 (84.8)	716 (80.8)	31,507 (83.9)	59 (81.9)	22,852 (85.1)
	1–7 days	223 (12.0)	3379 (8.7)	1091 (10.1)	2519 (8.3)	97 (11.0)	3279 (8.7)	8 (11.1)	2201 (8.2)
	≥1 week	237 (12.8)	2800 (7.2)	958 (8.9)	2085 (6.9)	73 (8.2)	2785 (7.4)	5 (6.9)	1801 (6.7)
Number of admissions, infection related [*n* (%)]	0	1705 (92.0)	37,587 (96.2)	10,196 (94.7)	29,111 (96.4)	845 (95.4)	36,069 (96.0)	69 (95.8)	25,929 (96.7)
	1	105 (5.7)	1202 (3.1)	456 (4.24)	854 (2.8)	29 (3.3)	1193 (32)	3 (4.2)	139 (2.8)
	≥2	43 (2.3)	294 (0.8)	115 (1.07)	224 (0.7)	12 (1.4)	309 (0.8)	0	186 (0.7)
Number of admission days, infection related [*n* (%)]	0	1714 (92.5)	37,715 (96.5)	10,246 (95.1)	29,198 (96.7)	852 (96.2)	36,194 (96.3)	-	-
	1–7 days	86 (4.6)	855 (2.2)	322 (3.0)	623 (2.1)	21 (2.4)	852 (2.3)	-	-
	≥1 week	53 (2.9)	513 (1.3)	199 (1.9)	368 (1.2)	13 (1.5)	525 (1.4)	-	-
Nursing home residency [*n* (%)]	Yes	42 (2.3)	807 (2.1)	249 (2.3)	604 (2.0)	18 (2.0)	781 (2.1)	-	-

* IQR = Interquartile range. ** For the different resistance patterns, the 33.3/66.6 percentiles were as following: mecillinam 10/21.88, trimethoprim 10/21.88, nitrofurantoin 10.5/22.00, multi-resistance 10/18 DDD. *** Numbers too few to show. **** If a patient was admitted and discharged from the hospital within the same day, but the admission was registered as a full-day admission, this counted as 1 admission, but as 0 days, hence the discrepancy between number of admissions and number of days admitted.

**Table 2 antibiotics-11-01382-t002:** Adjusted analyses of total exposure to antibiotics (number of prescriptions and DDD) and time since last prescription, and OR of resistant *E. coli*.

		Mecillinam		Trimethoprim		Nitrofurantoin		Multi-Resistance	
		OR *	CI95% **	OR	CI95%	OR	CI95%	OR	CI95%
Number of prescriptions	0	Ref		Ref		Ref		Ref	
	1	1.32	(1.15;1.53)	1.40	(1.32;1.5)	1.25	(0.97;1.61)	1.35	(0.59;3.08)
	2	1.82	(1.59;2.07)	1.66	(1.56;1.77)	2.10	(1.69;2.6)	2.86	(1.45;5.63)
	≥3	2.47	(2.18;2.8)	3.11	(2.92;3.31)	4.53	(3.76;5.46)	11.53	(6.37;20.85)
Number of DDD ***	0	Ref		Ref		Ref		Ref	
	>0–33.3 percentile	1.56	(1.39;1.76)	1.54	(1.46;1.63)	1.84	(1.51;2.23)	1.34	(0.59;3.05)
	>33.3–6.66 percentile	1.88	(1.63;2.18)	2.03	(1.89;2.17)	2.59	(2.07;3.24)	2.29	(1.07;4.95)
	>66.6 percentile	2.60	(2.26;2.99)	3.22	(2.99;3.46)	4.49	(3.65;5.51)	10.21	(5.78;18.02)
Number of prescriptions	0	Ref		Ref				Ref	
	1	2.29	(2.06;2.55)	2.23	(2.12;2.35)	2.95	(2.48;3.5)	6.23	(3.65;10.64)
	2	1.64	(1.42;1.9)	1.73	(1.62;1.86)	2.61	(2.08;3.27)	4.04	(2.1;7.77)
	≥3	1.60	(1.35;1.89)	1.51	(1.39;1.63)	1.87	(1.4;2.48)	2.32	(0.92;5.86)

* OR = Odds Ratio. ** CI95% = 95% Confidence interval. *** For the different resistance patterns, the 33.3/66.6 percentiles were a s following: mecillinam 10/21.88, trimethoprim 10/21.88, nitrofurantoin 10.5/22.00, multiresistance 10/18 DDD.

**Table 3 antibiotics-11-01382-t003:** Results of adjusted analysis of exposure to specific drugs and odds of resistance.

		Mecillinam		Trimethoprim		Nitrofurantoin		Multi-Resistance	
		OR *	CI95% **	OR	CI95%	OR	CI95%	OR	CI95%
Exposure to specific drugs within 8–90 days (reference: no exposure)									
Phenoxymethylpenicillin or dicloxacillin ***	Yes	2.01	(1.75;2.31)	1.18	(1.09;1.28)	1.28	(1;1.64)	1.06	(0.45;2.49)
Pivmecillinam	Yes	1.62	(1.46;1.8)	1.36	(1.29;1.43)	1.37	(1.16;1.63)	3.2	(2.07;4.95)
Amoxicillin	Yes	1.8	(1.43;2.27)	1.25	(1.09;1.42)	0.84	(0.49;1.44)	-	-
Amoxicillin + β lactamase inhibitor	Yes	1.75	(1.25;2.47)	1.31	(1.08;1.59)	1.48	(0.84;2.59)	10.18	(4.27;24.27)
Trimethoprim	Yes	1.43	(1.19;1.71)	6.48	(5.93;7.09)	1.05	(0.71;1.55)	8.67	(4.62;16.26)
Sulfamethizole	Yes	1.25	(1.07;1.47)	1.41	(1.31;1.53)	1.03	(0.78;1.35)	1.45	(0.74;2.83)
Nitrofurantoin	Yes	1.56	(1.32;1.84)	1.85	(1.7;2.02)	8.64	(7.26;10.29)	9.11	(5.47;15.16)
Macrolides	Yes	0.87	(0.64;1.18)	1.18	(1.03;1.34)	1.09	(0.71;1.68)	-	-
Quinolones	Yes	0.92	(0.69;1.22)	2.14	(1.91;2.4)	1.68	(1.16;2.43)	2.54	(0.86;7.52)
Others	Yes	1.76	(1.48;2.08)	2.25	(2.01;2.51)	1.21	(0.87;1.69)	3.16	(1.66;6.01)

* OR = Odds Ratio. ** CI95% = 95% Confidence interval. *** Phenoxymethylpenicillin and dicloxacillin were grouped together as narrow-spectrum penicillins.

## Data Availability

Due to Danish data protection law, the data applied in this study is not publicly available.

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
