# Peer review of "Prior Antibiotic Use Increases Risk of Urinary Tract Infections Caused by Resistant Escherichia coli among Elderly in Primary Care: A Case-Control Study"

_antibiotics, 2022, doi:10.3390/antibiotics11101382_

Round 1

Reviewer 1 Report

The authors have tried to demonstrate the prior use of antibiotic as a risk factor for the emergence of MDR E. coli.

Specific comments

Titles-

The title should be clear and simple as well as straight forward. It should reflect study setting and study design. 

Name of the bacteria should be italized then and there required

Since the MDR is 0.3 percent, what is the relevance of this study

Abstract

Overall, it is ok. 

But the study period is too older. Hence, the authors must include the information up to 2021.

What is the selection criteria for E. coli, since UTI can be caused by different bacteria.

Authors have to do the stereotyping of uro-pathogenic E. coli in order to make it more relevant

Inclusion and exclusion criteria are not given

It is better to include the reference of Kass count

Inferential statistics has to be improved

Results and discussions

It has to be refined

Author Response

REFEREE 1:

We thank the reviewer for the time spent reviewing our manuscript. We have addressed the issues raised below.

Title

The title should be clear and simple as well as straight forward. It should reflect study setting and study design                        

We have changed the title from:

“Prior antibiotic use: a risk factor for urinary tract infections caused by resistant Escherichia coli in the elderly population”

To:

“Prior antibiotic use increases risk of urinary tract infections caused by resistant Escherichia coli among elderly in primary care. A case-control study”

“Name of the bacteria should be italized then and there required

We have italicized all bacteria names. 

Since the MDR is 0.3 percent, what is the relevance of this study

We have analysed four outcomes in this study. Firstly, we examined resistance for mecillinam, trimethoprim or nitrofurantoin for each of these three agents separately. The resistance rates for the three agents ranged from 2.3% to 26.3%. As a fourth outcome, we analysed multiresistance defined as resistance to all three agents simultaneously. Although the rate of this resistance pattern is low, we judge this outcome highly clinical relevant, as there are few treatment options left.

The novelty of this study is especially the exploration of how prior use of antibiotics affects the risk of mecillinam resistance. To our knowledge, this has not been studied using individual level data before.

Abstract:

But the study period is too older. Hence, the authors must include the information up to 2021

Antimicrobial resistance is an increasing problem worldwide and the time period itself is not of major importance to the research question.

The time period selected for the present study has the advantage that the resistance rates for the included agents (outcomes) were relatively stable

What is the selection criteria for E. coli, since UTI can be caused by different bacteria

We decided to only include E. coli as this is by far the most common pathogen causing UTI in general practice.

We acknowledge that UTI can be caused by a large variety of pathogens other than E. coli and that this choice affects the external validity of the study to only include individuals infected with this specific pathogen. However, it also simplifies the explored associations. Including a wider range of pathogens would complicate the set up significantly, due to some commonly pathogens being intrinsic resistant to some of the agents included in this study. Furthermore, international susceptibility breakpoints do not exist for all  included agents for all uropathogens.

Lastly, we do acknowledge that when only including E. coli we are not able to take shifts to pathogens with intrinsic resistance into consideration.

We have elaborated on this further in the discussion in p11 line 1-6 to clarify the reasons behind our choices.

Further limitations of the study include that we chose to only include UTI episodes caused by E. coli. UTI can be caused by a variety of different pathogens, but E. coli is responsible for the vast majority of all UTI among elderly. We did not include all potential uropathogens, as it is difficult to address intrinsic resistance in this study design and there are no internationally recognized breakpoints for all potential uropathogens. Thus, the external validity of the study is limited to include individuals infected with E. coli.

Authors have to do the stereotyping of uro-pathogenic E. coli in order to make it more relevant

We have assumed that the reviewer is referring to serotyping.

As this is a large retrospective register study using routine data from microbiology reports, we do not have the possibility to perform additional laboratory analysis on the cultured E. coli isolates. We recognize that serotyping (with regard to clonality) or any further characterization of especially the resistant E. coli isolates would be interesting, but this would be beyond the scope of this study.

We have included a sentence in the discussion regarding this at p10 line 47-49.

“Because of the retrospective nature of the study design using routine data from microbiology reports, we do not have information on clonality or other characterization of the resistant E. coli isolates.”

Inclusion and exclusion criteria are not given

We have included the following paragraph to clarify the inclusion and exclusion criteria in the method section at p2 line 28-30.

“The study only included urine samples with significant growth of E. coli and susceptibility testing to one or more of the following antibiotics: mecillinam, nitrofurantoin or trimethoprim”

It is better to include the reference of Kass count.

Kass proposed a threshold of 100.000 CFU/ml for significant bacteriuria. However, in our laboratories, the revised threshold of 1.000 CFU/ml for significant bacteriuria with E. coli is used, as recommended in European guidelines (1).

Inferential statistics has to be improved.

We can interpret the reviewers comment in two ways:

a)     The statistical methods applied (the statistical test and the corresponding measure of the association, in our case the OR) are inefficient.

Due to the large study population inference will be strong. Furthermore, the OR is the natural choice of effect measure in a case control study.

b)     The conclusions drawn from the statistics are not written clearly. 

We have re-read the result section and enhanced the clarity and readability of the reported results.

Results and discussions

It has to be refined.

We have revised the result and discussion sections to improve their clarity.

  • European guidelines for urinalysis: a collaborative document produced by European clinical microbiologists and clinical chemists under ECLM in collaboration with ESCMID. Aspevall O, Hallander H, Gant V, Kouri T. Clin Microbiol Infect. 2001 Apr;7(4):173-8. doi: 10.1046/j.1198-743x.2001.00237.x

Reviewer 2 Report

We commend your focus on E. coli resistance related to prior antibiotic use in the elderly, outpatient population. However, the present manuscript is quite confusing in its methods and lacks clear definitions. It is also missing data for the sensitivity analysis but still makes claims about what was found. Further, claims about pivmecillinam are made in this paper are not supported by data as the study was not designed for this. Some comments for consideration prior to resubmission are below:

Methods:

·       Please clarify definition of UTI episode: If looking at antibiotics within 90 days of a urine culture but only including episodes with a full 90 days in between, does this mean you are only including antibiotic use for infections other than UTI within the preceding 90 days?

·       A minimum of 1,000 CFU was used to define a UTI. Is this the standard in Denmark? Is the threshold lower to include catheter-associated UTIs? Please further comment on types of UTIs included within the study. If Denmark follows the European Association of Urology Guidelines, consider using their definitions located in section 3.3.3: https://uroweb.org/guidelines/urological-infections/chapter/the-guideline

·       The case and control definition is not entirely clear as the separation of drugs with slashes does not tell me if we are including resistance to one drug OR the other, or resistance to the first AND second drug. Further, total numbers of cases and controls is not mentioned. I would recommend at minimum stating these numbers in Table 1.

·       Recommend excluding hospital admissions and any patient’s hospitalized within their “UTI episode.” You mention the lack of information from hospitalizations to be a limitation of your study, and this lack of information has the potential to introduce many confounders. Since the goal of your study is outpatients, I would recommend sticking to patients that were outpatient for the duration of their inclusion in the study.

·       What is the standard definition of a nursing home in Denmark? Is it uncommon for both sets of parents/grandparents to live in a home with their married/partnered children? Recommend further clarifying this definition.

 Results

·       Consider breaking up Table 1, potentially at exposure within 90 days.

·       Further clarify DDD.

·       Section 3.3 could use rephrasing for clarity. Sentence 2 for example could read more similarly, “The OR for mecillinam resistance increased from 1.51 with no exposure in 90 days to 2.13 with exposure 61-90 days prior to the UTI episode.”

·       Figure 2 does not necessarily add anything to the paper and is confusing. It lack axis labels and I am unsure how to interpret this figure at present.

·       Table 3: Why group together phenoxymethylpenicillin and dicloxacillin? Recommend explaining in table foot notes at minimum.

·       Section 3.5 Sensitivity Analysis is incomplete. Statements cannot be made without accompanying data.

Discussion

·       Overall, can be condensed for brevity.

·       Would include potential inclusion of asymptomatic bacteria as a limitation of the study.

·       Missing animal husbandry as a potential limitation- introduces resistance into the population

·       You report that pivmecillinam is a safe option in your discussions and conclusions; however, the study is not designed to compare the safety or efficacy of the agents frequently used in UTI. Therefore, this claim is inappropriate.

·       No attempt is made to explain why there may be less resistance with pivmecillinam.

·       There was a paper published within Antibiotics in May of this year with a similar design in a German cohort: https://www.mdpi.com/2079-6382/11/6/751 Would recommend making mention of this study within your discussion.

Supplemental Materials

·       Please list out the descriptors of the ICD-10 codes as well.

Author Response

ESTRA REFEREE

Generelt

We commend your focus on E. coli resistance related to prior antibiotic use in the elderly, outpatient population. However, the present manuscript is quite confusing in its methods and lacks clear definitions. It is also missing data for the sensitivity analysis but still makes claims about what was found. Further, claims about pivmecillinam are made in this paper are not supported by data as the study was not designed for this. Some comments for consideration prior to resubmission are below

Thank you for this overall positive assessment regarding the relevance of our study.

We have addressed the concrete comments below point-by-point.

Methods

Please clarify definition of UTI episode: If looking at antibiotics within 90 days of a urine culture but only including episodes with a full 90 days in between, does this mean you are only including antibiotic use for infections other than UTI within the preceding 90 days?

In Denmark In-house microbiological testing and susceptibility testing of urine, are very common in general practice. UTI is therefore often diagnosed and treated with antibiotic without a urine sample being sent to the department of clinical microbiology. All these UTI treatments are therefore captured as exposure.

A minimum of 1,000 CFU was used to define a UTI. Is this the standard in Denmark? Is the threshold lower to include catheter-associated UTIs? Please further comment on types of UTIs included within the study. If Denmark follows the European Association of Urology Guidelines, consider using their definitions located in section 3.3.3: https://uroweb.org/guidelines/urological-infections/chapter/the-guideline

Yes, the threshold for significant bacteriuria with E. coli is 1,000 in Danish laboratories in accordance with the European guidelines. It is the same for all non-sterile urine samples, thus including all relevant sample types taken in general practice. We have clarified that urine samples mainly consist of midstream urine samples and urine sampled via catheter urine. (p3, line 3-4):

“Data on the urine samples (mainly midstream urine samples and urine sampled via catheter) were extracted from the two DCM’s laboratory information systems.”

Thank you for your attention to the EAU-guidelines. However, as these refer to clinical entities, we have not included these, as we only have very limited clinical information. The mentioned section 3.3.3 refers to asymptomatic bacteriuria which is not included in our study, where the case definition includes antibiotic treatment. We have instead included the European guidelines as a reference in the Material and Methods section, where the laboratory methods are now described in more detail (p3, line 18-27)

“Urinalysis was performed according to European guidelines. Urine specimen were plated on split plates (5% horse blood and a chromogenic UTI-agar) and read after overnight in-cubation at 35±1ºC in atmospheric air. Significant growth of E. coli was defined as mini-mum 1,000 CFU/ml. Bacterial species identification was based on phenotypical and bio-chemical characteristics and possibly confirmed using Matrix-Assisted Laser Desorp-tion-Ionization time of flight mass spectrometry. Susceptibility testing was performed us-ing EUCAST disk diffusion methodology and breakpoints [23]. For trimethoprim, the breakpoints implemented per 01-01-2020 (R <15 mm to S ≥15 mm) were applied throughout the study period, meaning that former “I=intermediate” was categorized as “S=susceptible”.”

The case and control definition is not entirely clear as the separation of drugs with slashes does not tell me if we are including resistance to one drug OR the other, or resistance to the first AND second drug. Further, total numbers of cases and controls is not mentioned. I would recommend at minimum stating these numbers in Table 1.

Thank you for pointing out the slashes being confusing. We have clarified this in method section (section 2.3) and changed the slashes to or in the text.

We have included a line in Table 1 with stating number of cases and controls. This line was unfortunately cut when the manuscript was set up in the Antibiotic format. We apologize for this.

Recommend excluding hospital admissions and any patient’s hospitalized within their “UTI episode.” You mention the lack of information from hospitalizations to be a limitation of your study, and this lack of information has the potential to introduce many confounders. Since the goal of your study is outpatients, I would recommend sticking to patients that were outpatient for the duration of their inclusion in the study.

Thank you for this suggestion. Hospitalization is a confounder in the sense that the lack of information on AB exposure in the hospital gives artificially lower AB exposure in the data in our study and may be related to higher resistance rates. This confounding is adjusted for by the variables measuring prior hospitalization in our analysis. Furthermore, due to admission to hospital being common in this population, such exclusion would exclude a considerable number of UTI episodes and therefore lower the external validity. For example, 27% of all cases would be excluded in the analysis of mecillinam resistance.

Note that all data is based on UTI suspicion in primary care, indicating that these people were not in the hospital at the moment of the index UTI episode.   

What is the standard definition of a nursing home in Denmark? Is it uncommon for both sets of parents/grandparents to live in a home with their married/partnered children? Recommend further clarifying this definition.

A nursing home is a common term for institutions where various degrees of care is provided.

It is very uncommon in Denmark, that ≥3 individuals aged 80 years live at the same address in a non-nursing home context.

Results

Consider breaking up Table 1, potentially at exposure within 90 days.

That you for this suggestion. We have discussed this among the authors and have decided to keep the table as one.

Further clarify DDD.

To further clarify the DDD-intervals applied in the analysis we have added the following sentence (p4, line 15-16)

“DDD was grouped as no exposure and three equally large groups of exposure constructed as follows: >0–33.3 percentile, >33.3–66.6 percentile and >66.6 percentile.” All specific numbers are provided as footnotes in the tables.

Section 3.3 could use rephrasing for clarity. Sentence 2 for example could read more similarly, “The OR for mecillinam resistance increased from 1.51 with no exposure in 90 days to 2.13 with exposure 61-90 days prior to the UTI episode.”

We have revised the result section.

Figure 2 does not necessarily add anything to the paper and is confusing. It lack axis labels and I am unsure how to interpret this figure at present.

We have replaced the figure with a table.

Table 3: Why group together phenoxymethylpenicillin and dicloxacillin? Recommend explaining in table foot notes at minimum.

The two agents were group together as narrow-spectrum penicillins. We have included a foot note in the tables stating that:

”Phenoxymethylpenicillin and dicloxacillin were grouped together as narrow-spectrum penicillins”

Section 3.5 Sensitivity Analysis is incomplete. Statements cannot be made without accompanying data.

We have included the results of the sensitivity analysis in the supplementary material.

Discussion

Overall, can be condensed for brevity.

We have revised and condensed the discussion

Would include potential inclusion of asymptomatic bacteria as a limitation of the study.

We have included a note on this in the discussion at p10, line 51-53

“However, some GPs may still treat ASB with antibiotics and we cannot exclude that some cases of ASB have been included. This is a limitation of the study.”

Missing animal husbandry as a potential limitation- introduces resistance into the population

Unfortunately, we did not have any data on husbandry nor travel history. We have included a sentence addressing this  in the discussion, p11 line 15-17.

“Finally, we do not have clinical information such as travel history and animal husbandry which affect resistance prevalence.“

You report that pivmecillinam is a safe option in your discussions and conclusions; however, the study is not designed to compare the safety or efficacy of the agents frequently used in UTI. Therefore, this claim is inappropriate.

Thank you for this insightful comment. It is absolutely right – the claim that mecillinam is a safe choice of treatment is unsupported by the present study. We have rephrased our claim, to state that the results of the study indicate that mecillinam is a favorable choice of treatment in terms of selection.

No attempt is made to explain why there may be less resistance with pivmecillinam.

In the discussion we reference Poulsen et al. and their study regarding clonal spread of mecillinam resistant E. coli. They found high genetic diversity among the mecillinam resistant pathogens indicating little clonal spread of mecillinam resistant E. coli. We did not explore this aspect further as it is not the main focus of the article.

There was a paper published within Antibiotics in May of this year with a similar design in a German cohort: https://www.mdpi.com/2079-6382/11/6/751 Would recommend making mention of this study within your discussion

Thank you for pointing out this publication for us. We have included it as reference in the discussion.

Supplementary materials

Please list out the descriptors of the ICD-10 codes as well.

We have listed the descriptors for the ICD-10 codes in the supplementary material.

Reviewer 3 Report

The manuscript by Maria Jensen et al. describes an interesting study on the impact of increased antibiotic use on E. coli in elderly populations.

This manuscript deserves to be fully reviewed before possible publication.

Global: italicize the names of bacteria, "ie.", "e.g.".

Prefer passive forms.

Numbers less than 12 should be written in capital letters.

Subheadings are incorrectly numbered.

Methods: why did the authors prefer the case-control model, urinary tract infections being frequent, a prospective/retrospective cohort would be more interesting, this is a major limitation!

Methods: patients included a second time are major biases, even with a sensitivity analysis, and the second/third/etc. sampling should be removed (not randomly selected). 

Why didn't the authors decide to analyze bi-antibiotic resistance?

Culture methodologies need to be more fully detailed.

Lines 140-148 should be fully written (not in bullet form, as seems to be the case now...).

Results: A flow chart is essential and necessary for readers to understand the selection process.

Table 1 is truncated.

The figure is difficult to read and does not have a title. 

Funding: What is the Velux Foundation (not the small window I presume...).

Author Response

REFEREE 2:

Global:

italicize the names of bacteria, "ie.", "e.g.".

We have italicized all bacterial names and Latin abbreviations.

Prefer passive forms

We have thoroughly revised the manuscript with regards to passive forms, where appropriate

Numbers less than 12 should be written in capital letters

We have revised the manuscript to change all number less than 12 to text if the number was not written as part of an interval for variables. 

Subheadings are incorrectly numbered

This has been corrected.

Methods:

why did the authors prefer the case-control model, urinary tract infections being frequent, a prospective/retrospective cohort would be more interesting, this is a major limitation!

It is correct that UTI in general is a very frequent infection – in Denmark it is in fact the most common indication for antibiotic treatment among elderly patients. Furthermore, the exposure of antibiotics is also common.

However, despite UTI and antibiotic treatment in general is very common, the outcome of this study (resistance) is not and hence the case control study is an obvious choice of design.

A prospective cohort design would sample all UTI suspicion cases, and then those for which an antibiotic was prescribed, and verifies the resistance pattern for each of these. Since only a fraction of UTI suspicion cases was followed by an urine sample send to the laboratory, tentatively those cases for which resistance was probable, the resulting sample, omitting the UTI suspicion cases for which no urine sample was send to the laboratory, is an artificially high prevalence of resistance sample. Therefore, in this sample, the raw prevalences are not generalizable, but, if analysed by logistic regression, i.e. as a case-control design, the effect measures, i.e. the ORs, are (as multiplicative increases in odds for resistance). Note that the above data is exactly the data we use here, and that the same logistic regression analysis would be the analysis method of choice even if we view the sample as a cohort design. Hence, whether we have a case-control design or a cohort design is a question of semantics.    

patients included a second time are major biases, even with a sensitivity analysis, and the second/third/etc. sampling should be removed (not randomly selected)

In the analysis we adjusted for repeated measures using a generalized estimating equation, and we will therefore argue that the inclusion of UTI episodes following the first episode strengthens the analysis.

To exclude all other samples than the first one that appears in our sample would, in turn, introduce a selection bias. Since only a fraction of UTI cases are followed by a urine sample send to the laboratory, the first record in our data does in general not coincide with the first UTI suspicion for that person and will have no special meaning.

If we only included the first sample, we would furthermore sample considerably more UTI episodes in the earlier years of the study period and from younger individuals.

We have performed a sensitivity analysis, where a single random UTI episode from each individual was selected. These results were similar to the main analysis and are now included in the supplementary material.

Why didn't the authors decide to analyze bi-antibiotic resistance?

As we already report four outcomes, we decided not to analyze bi-antibiotic resistance to keep the result section simpler and clear. Our assessment is, that the interaction of resistance to different agents is sufficiently covered in the analysis for multi-drug resistance.

Culture methodologies need to be more fully detailed

We have elaborated more fully on the culture and susceptibility testing in the method section at p3 line 19-26

“Urinalysis was performed according to European guidelines. Urine specimen were plated on split plates (5% horse blood and a chromogenic UTI-agar) and read after overnight incubation at 35±1ºC in atmospheric air. Significant growth of E. coli was defined as minimum 1,000 CFU/ml. Bacterial species identification was based on phenotypical and biochemical characteristics and possibly  confirmed using Matrix-Assisted Laser Desorption-Ionization time of flight mass spectrometry. Susceptibility testing was performed using EUCAST disk diffusion methodology and breakpoints [23]. For trimethoprim, the breakpoints implemented per 01-01-2020 (R <15 mm to S ≥15 mm) were applied throughout the study period, meaning that former “I=intermediate” was categorized as “S=susceptible”.”

Lines 140-148 should be fully written (not in bullet form, as seems to be the case now...).

We have rewritten this section as a coherent text instead. Please see p4, line 24-40

Results:

A flow chart is essential and necessary for readers to understand the selection process

We have included a flow chart in the article to help clarify the selection process.

Table 1 is truncated.

We have inserted the missing row. Thank you for pointing this out.

The figure is difficult to read and does not have a title

We have replaced the figure with a table.

Funding:

What is the Velux Foundation (not the small window I presume...)

The Velux foundation is a non-profit foundation founded 1981 by Mr. V. Rasmussen – the founder of window company Velux A/S

Reviewer 4 Report

The study is very interesting and shows important results in terms of clinical utility.

I have only minor comments in order to improve the discussion:

- what about the role of fosfomycin? Fosfomycin is one of the most prescribed and used ABT.

- what about the role of cross-resistances in determining treatment failure?

Finally, please consider the following manuscript for discuss the role of fosfomycin in UTI: Int J Antimicrob Agents. 2022 May;59(5):106574. doi: 10.1016/j.ijantimicag.2022.106574. Epub 2022 Mar 18.

Author Response

1. The study is very interesting and shows important results in terms of clinical utility.I have only minor comments in order to improve the discussion:

Thank you for this positive assessment.

2. what about the role of fosfomycin? Fosfomycin is one of the most prescribed and used ABT

In Denmark, fosfomycintrometamol has not been available until 2019. We have now clarified this in the Materials and Methods section  where fosfomycin was deleted:

“UTI treatments were defined as prescription of the following antibiotics available for UTI treatment in Denmark: pivmecillinam, trimethoprim, nitrofurantoin, sulfamethizole, ciprofloxacin, pivampicillin and amoxicillin.”

3. what about the role of cross-resistances in determining treatment failure?

We have only included the period 8-90 days prior to urine sampling in the exposure period to exclude treatment failure. Therefore, we do believe that treatment failure is not captured in this study. Please se p4, line 10-12

4. Finally, please consider the following manuscript for discuss the role of fosfomycin in UTI: Int J Antimicrob Agents. 2022 May;59(5):106574. doi: 10.1016/j.ijantimicag.2022.106574. Epub 2022 Mar 18

Thank you for this suggestion. However, as fosfomycin has not been available in Denmark during the study period and is not further addressed in the manuscript, we have decided not to include this article as a reference.

Round 2

Reviewer 2 Report

None

Reviewer 3 Report

The authors have revised their manuscript that is now suitable for publication.